# The Efficient and Convenient Extracting Uranium from Water by a Uranyl-Ion Affine Microgel Container

**DOI:** 10.3390/nano12132259

**Published:** 2022-06-30

**Authors:** Peiyan He, Minghao Shen, Wanli Xie, Yue Ma, Jianming Pan

**Affiliations:** School of Chemistry and Chemical Engineering, Jiangsu University, Zhenjiang 212013, China; xiakexu1020@163.com (P.H.); shenminghao1996@163.com (M.S.); wanlixie3719@aliyun.com (W.X.); yamma@ujs.edu.cn (Y.M.)

**Keywords:** microgel, self-assembly, adsorption

## Abstract

Uranium is an indispensable part of the nuclear industry that benefits us, but its consequent pollution of water bodies also makes a far-reaching impact on human society. The rapid, efficient and convenient extraction of uranium from water is to be a top priority. Thanks to the super hydrophilic and fast adsorption rate of microgel, it has been the ideal adsorbent in water; however, it was too difficult to recover the microgel after adsorption, which limited its practical applications. Here, we developed a uranyl-ion affine and recyclable microgel container that has not only the rapid swelling rate of microgel particles but also allows the detection of the adsorption saturation process by the naked eye.

## 1. Introduction

Nuclear energy is a proven source of power, which provides huge and sustained electricity with ultra-high energy density and ultra-low greenhouse gas emissions [1,2,3]. As predicted by the International Atomic Energy Agency, nuclear power may overtake oil as the dominant energy source for decades to come [4,5,6]. Uranium is one of the most widely used raw materials in the nuclear industry that does not produce greenhouse gases [7,8]. However, uranium pollution, especially in water, caused by nuclear waste and leakage accidents, is unavoidable [9,10]. The adsorption and recovery of uranium in water, consequently, are a top priority [11].

In order to extract uranium from water, the commonly used methods are ion exchange, chemical precipitation, coagulation and adsorption [12,13,14,15,16]. However, the first three methods have inevitable defects, for example the ion exchange method is faced with low extraction capacity and a complicated regeneration process; the chemical precipitation method often causes secondary pollution; and the coagulation method is accompanied by poor selectivity [17,18]. Fortunately, an adsorption method involving extracting uranium by designable adsorbents, shows high adsorption efficiency and selectivity through the modification of special groups of adsorbents [19,20]. Furthermore, in order to elute and recover uranium from other impurities, the adsorption method is undoubtedly the most facile [21]. However, the synthesis of traditional adsorbents is usually complex, including the processes of structure-formation, pore-creation and functional group-modification [22]. The common materials used for uranium absorption have been obtained by modifying uranium affinity groups on the surface of relatively hard particles [23,24]. This costly step-by-step synthesis process leads to the masking of internal sites, which reduces the adsorption efficiency and, as such, the small particles are difficult to gather, which is also a common problem of uranium-recycling [25]. Furthermore, most adsorbents are composed of an organic skeleton, which greatly reduces their water compatibility and limits their adsorption efficiency and selectivity in aqueous solution [26]. For uranium ions in water, the hydrophobic materials do not seem to be the ideal absorbents [27,28]. Therefore, it is essential to develop a simple strategy to obtain water compatible adsorbents for selective uranium adsorption [29].

In recent years, hydrophilic polymer materials have received increasing attention in the field of water treatment due to their efficient adsorption properties for metal contaminants in water, especially the cross-linked polymer hydrogels with adsorption, water absorption, water retention and slow release functions [30,31,32]. Since the swelling rate of hydrogel is inversely proportional to volume, obviously that of a microgel is higher than that of the general macro-hydrogel [33,34]. With the advantage in swelling rate, microgel displays a high adsorption rate and shows great potential in emergency water treatment [35]. However, the microgel with a hydrated molecular chain and a strong charge is too small to gather and take out after adsorption, and therefore cannot really achieve the removal of pollutants [36]. Therefore, in this work, a novel microgel container (micro-container) with uranium affinity was developed in order to obtain both high swelling rate and convenience of recycling, which is to innovatively expend the strategies of uranium extraction [37,38].

According to our design, the copolymerization of *N*-isopropylacrylamide (NIPAAm), acrylamide (AAm) and *N*,*N*-Methylenebisacrylamide (MBAAm) was to create the backbone of the microgel, and vinyl phosphate (VPA) was employed to generate the molecular affinity towards uranyl ions. The phosphate functionalized microgel (VPA-microgel) composed of the above monomers was prepared via surfactant-assisted emulsion precipitation polymerization reported by our group previously [39]. In order to self-assemble the parts into a whole, another step was carried out to prepare a microgel container that uses Aam and a cross-linker as a macromolecular interpenetrating network after UV light-initiated polymerization in the presence of VPA-microgel. As the property of the colloidal photonic crystal of a PNIPAAm based microgel, the microgel container showed corresponding optical properties. Finally, specific trapping properties, such as their binding kinetics and equilibrium, were studied and discussed in detail. Compared with other uranium absorbents, the micro-container is low-cost and environmentally friendly and, especially in this study, the functional groups can be changed flexibly to combine with other chemicals. Moreover, we believe that the work may provide a paradigm for the fabrication of adsorbents for extracting seawater substances and treating water pollution.

## 2. Materials and Methods

### 2.1. Chemicals

*N*-Isopropylacrylamide (NIPAAm, 98%), sodium dodecyl sulfate (SDS, 99.5%), Vinylphosphoric acid (VPA, 95%) and *N*,*N*-Methylenebisacrylamide (MBAAm, 99%) were purchased from Aladdin Industrial Corporation (Shanghai, China) and Uranyl Nitrate Hexahydrate (99%) was purchased from Macklin Biochemical Technology Co. (Shanghai, China). All reagents were analytically pure and were not further purified prior to use. The experimental water was double distilled water.

### 2.2. Instrumentation

The morphology and structures of the prepared materials were observed by scanning electron microscope (SEM, JEOL, JSM-7800, Japan). FT-IR spectra (4000–400 cm^−1^) were recorded on Nicolet NEXUS-470 FTIR apparatus (USA) using KBr disks. Surface elements were investigated by X-ray photoelectron spectroscopy (XPS, ESCALAB 250, Apreo S HiVac, Thermo Scientific).

### 2.3. Preparation of Phosphate Functionalized VPA-Microgel

The synthesis of VPA-microgel was presented as follows: NIPAAm (580 mg), BIS (22 mg), VPA (30 mg) and SDS (6 mg) were added into a three-neck round bottom flask (100 mL) and then 50 mL of deionized water was put into the flask. After ultrasonication, a clear solution was obtained and purged N_2_ for 30 min at room temperature in order to remove O_2_. The flask was immersed into a thermostatic water bath solution and was heated to 70 °C, and then KPS solution (40 mg dissolved in 2.5 mL of deionized water) was dropped to initiate the reaction. The polymerization lasted for 5 h at 70 °C with N_2_ conditions. After the reaction, the white suspension was purified by dialysis against water with frequent water change for 2 weeks, lyophilized, and stored in the refrigerator for further use.

### 2.4. Preparation of Micro-Container

The synthesis of the micro-container was performed as follows: photoinitiator HMP (6.6 mg), acrylamide (20 mg), crosslinker BIS (20 mg) and deionized water (1 m) were added into a Centrifugal tube 1 mL of deionized water. A clear solution was obtained after ultrasonication and VPA-microgel (50 mg) was transferred to the above monomer solution. After swelling in the dark for 12 h, a green colloidal crystal was obtained, which was then transferred into a 10 × 10 × 3 mm mold. The system was then triggered to polymerize by exposure to UV light at room temperature for 24 h (λ = 365 nm). The resulting films were released from the device and dried at 30 °C for further use.

### 2.5. Static Adsorption of Micro-Container

#### 2.5.1. Hydrogen Ion Concentration Optimization

In this part, the effect of the binding ability between phosphate group and uranyl ion in the acidic solutions with different H^+^ concentration was investigated by regulating the nitric acid content. The concentration of uranium ions (UO_2_^2+^, 100 mg L^−1^) was kept constant, and the test solution was prepared with different hydrogen ion concentrations (pH 1~6) by adding nitric acid. Different solutions and 10 mg of dried VPA-hydrogel were mixed and incubated at 293 K, respectively. After 2.0 h, the upper clear solution was filtered using a filter and the concentration of uranium was measured by inductively coupled plasma spectrometry emission (ICP) and was repeated three times. The equilibrium adsorption amount (*Q_e_*, mg g^−1^) was calculated by Equation (1):(1)$$Q_{e}=\frac{\left(C_{0}-C_{e}\right)V}{mM},$$
where *Q_e_* (mg/g) denotes the equilibrium adsorption capacity; *C*_0_ (mg L^−1^) denotes the original concentration of UO_2_^2+^ in the solution; *C_e_* (mg L^−1^) denotes the concentration of UO_2_^2+^ in the solution at equilibrium; *V* (mL) denotes the volume of the adsorption solution; *m* (g) denotes the mass of the adsorbent micro-container and *M* denotes the relative molecular mass of UO_2_^2+^ (*M* = 270.03).

#### 2.5.2. Adsorption Kinetic

In this experiment, the adsorption kinetic of the micro-container was investigated. The experimental procedure was as follows: 10 mg of dried micro-container was weighed precisely and placed in a 10 mL centrifuge tube. Then, 10 mL of the initial solution with a UO_2_^2+^ concentration of 500 mg L^−1^ was added into the centrifuge tube. Fourteen of the above tubes were sealed and subjected to static adsorption in a water bath shaker at 293 K. The upper clear layer was taken at different time intervals and the clear solution was treated according to the above experiment. The adsorption quantity (*Q_t_*, mg/g) was calculated as shown in Equation (2):(2)$$Q_{t}=\frac{\left(C_{0}-C_{t}\right)V}{mM},$$
where *Q_t_* (mg g^−1^) denotes the adsorption capacity at time *t*, *C*_0_ (mg L^−1^) denotes the original concentration of UO_2_^2+^ in the solution; *C_e_* (mg L^−1^) denotes the concentration of UO_2_^2+^ in the solution at equilibrium; *V* (mL) denotes the volume of the adsorption solution; *m* (g) denotes the mass of the adsorbent micro-container and *M* denotes the relative molecular mass of UO_2_^2+^.

#### 2.5.3. Equilibrium Adsorption

In order to measure the equilibrium adsorption of the micro-container, 10 mg of dry hydrogel was mixed with UO_2_^2+^ solution at different concentrations, respectively. The process of static adsorption and the measurement of uranium concentration were consistent with the previous test. The obtained data of IPC result were used to calculate the equilibrium adsorption amount (*Q_e_*, mg g^−1^), which was calculated in Equation (1).

#### 2.5.4. Reflectance Spectral Change Test before and after Adsorption

In order to study the color change of the micro-container before and after adsorption, its reflection spectra were tested. Two dried samples of micro-container were put into two 10 mL centrifuge tubes, and then 10 mL of UO_2_^2+^ solution with a concentration of 500 mg·L^−1^ at pH 3 was added into the two centrifuge tubes, which were sealed and subjected to static adsorption at 293 K in a water bath shaker. After 2 h, the two block hydrogels after binding were taken out separately and their reflection wavelengths were measured by fiber optic spectrometer, and the data were recorded. Another two samples were incubated in the same solution but without the UO_2_^2+^, and were subjected to the same process; the data were recorded as before the adsorption.

## 3. Results and Discussion

According to our previous method, VPA-containing microgel (VPA-microgel) was synthesized in aqueous solution through surfactant-assistant precipitation polymerization, using sodium dodecyl sulfate (SDS), potassium persulfate (KPS), Vinylphosphoric acid (VPA), N-isopropylacrylamide (NIPAAm), and methylene bisacrylamide (BIS) as the surfactant, initiator, phosphate functional monomer, backbone monomer, and crosslinker, respectively (Figure 1). SDS was chosen because it could stabilize particles during polymerization, while VPA was selected as it could provide a phosphate group that could interact with the uranyl group. To obtain the microgel container (micro-container), the resultant VPA-microgel was first co-incubated with the phosphate functional monomer, crosslinker and water-soluble photoinitiator (2-hydroxyethoxy)-2-methylpropiophenone (HMP) in a 10 × 10 × 3 mm mold. Then, a UV light-initiated polymerization was carried out after the VPA-microgel swelled completely and showed a clear blue color. Finally, the microgels formed a container-like rectangular hydrogel, due to the phosphate monomer and crosslinker copolymerized to construct the macromolecule interpenetrating network in the microgels’ gap. Due to the existence of phosphate groups and the water compatibility of the microstructure, it was anticipated that the micro-container would show a strong ability to adsorb and separate uranium rapidly from the water.

### 3.1. Morphology and Composition

The size and morphology of VPA-microgel were analyzed by dynamic light scattering (DLS) and scanning electron microscopy (SEM), respectively. As shown in Figure 2a, the average diameter of VPA-microgel was 610 nm with a uniform size distribution with a PDI of 0.026. The SEM image (Figure 2b) showed that all the particles exhibited a spherical shape. However, the particle size of the VPA-microgel reflected by SEM was only about 100 nm, which was much smaller than that obtained by DLS, probably due to the different hydration states under the two characterization conditions (wet vs. dry).

Chemical component of the VPA-microgel was further characterized using infrared spectroscopy (IR). Figure 3a displays the peaks of several corresponding groups. A broad absorption peak in the range of 3200~3400 cm^−1^ corresponds to N-H stretching vibrations. The multiplicative contraction vibration peak at 3077 cm^−1^ was attributed to the second strongest amide band. The C-H vibration peaks of methyl and hypomethyl at 2973, 2935, 2875 cm^−1^ were also observed. The contraction vibration peak of C-N near 1548 cm^−1^ and the bending vibration peak of -CH_3_ near 1459 cm^−1^ were displayed. The peaks formed by the symmetric vibrational coupling splitting of the double methyl group on -CH(CH_3_)_2_ were presented near 1375 cm^−1^ [40,41,42]. All the results of infrared spectroscopy proved the existence of the main components of the VPA-microgel. However, the characteristic peak of the phosphate group was not observed in the IR spectrum; another characterization method (XPS) was used to support proof.

The X-ray photoelectron spectroscopy (XPS) spectra of the VPA-microgel are shown in Figure 3b, showing that the characteristic peaks at 134.2 eV, 284.7 eV, 398.9 eV and 531.6 eV were attributed to the corresponding P 2p, C 1s, N 1s and O 1s electrons, respectively. The characteristic peak of P 2p clearly indicated that VPA-microgel contained phosphate groups which could adsorb uranyl ions specifically [43,44]. These above results indicated the successful preparation of monodispersed and phosphate functional VPA-microgel.

The VPA-microgel was then swollen and assembled into an ordered structure (observed at Figure 2c) in aqueous solution with a phosphate functional monomer and crosslinker, followed by UV light-initiated polymerization (Figure 1b). After forming a block hydrogel with a macromolecule interpenetrating network, the micro morphology of the micro-container is shown in Figure 2c,d; an obvious particle (diameter around 100 nm) structure can be observed in the dried hydrogel matrix, which demonstrates that our strategy successfully prepared a container-like hydrogel composed of microgels. However, the particle size was not as uniform as that of the VPA-microgel due to the deviation in the depth embedded in the matrix. Fortunately, the SEM photos showed a somewhat short-range ordered structure of the embedded particles, which means that the micro-container must have a colloidal crystal structure formed by the VPA-microgel. Undoubtedly, the IR and XPS spectra showed similar curves to the corresponding characterization curves of the VPA-microgel. These results together demonstrated that our micro-container design could be used as a general method for assembling the microgel parts into a whole hydrogel with complementary advantages.

### 3.2. Adsorption Experiment

As expected, the hydrophilic micro-container with phosphate groups could adsorb and separate uranium from an aqueous solution, since acidity is one of the most important parameters that influences the interaction between phosphate group and uranyl ions. In order to achieve the optimized adsorption capacity of the micro-container, the adsorptions of uranium at different H^+^ concentrations were carried out through incubating the micro-container in uranium ion solutions with different acidities. Moreover, phosphate group exists stably only in an acidic environment—it was not investigated in an alkaline environment. The results are displayed in Figure 4, showing that the adsorption capacity of the micro-container firstly increased and then decreased with the increase of acidity. It is worth noting that the maximum value was reached at pH 3 (H^+^ concentration was 10^−3^ M), and so all the other adsorption experiments were performed at pH 3.

The binding kinetics of the adsorbate uranium with the micro-container was evaluated by batch adsorption experiments, incubating the adsorbent in fixed concentrations of uranyl ions for time variables. The adsorption capacity of the micro-container was examined and counted at different time points to obtain the adsorption kinetic curves. Figure 5 shows that the micro-container reached its binding equilibrium at a time of about 2 h, demonstrating quite fast binding processes. As known, the adsorption kinetic model of the micro-container was confirmed by the diffusion-controlling step. So, in order to further elucidate the template binding rate and potential rate-controlling step in the adsorption process, the adsorbate binding kinetic data of the micro-container was fitted with two kinetic models, which are the pseudo-first-order model and the pseudo-second-order model, and all the details are listed in Table 1. The pseudo-first-order and pseudo-second-order kinetic equations are shown in Equations (3) and (4), respectively.
(3)$$Q_{t}=Q_{e}-Q_{e}\cdot e^{-k_{1}t}$$
(4)$$Q_{t\;}=\frac{k_{2}Q_{e}^{2}t}{1+k_{2}Q_{e}t}$$
(5)$$h=k_{2}Q_{e}^{2}$$
(6)$$t_{1/2\;}=\frac{1}{k_{2}Q_{e}\;},$$
where *k*_1_ (L/min) is the rate constant for the pseudo-first-order kinetic adsorption equation and *k*_2_ (g/(mg·min)) is the rate constant for the pseudo-second-order kinetic adsorption equation. *h* (mg/(g·min)) represents the original adsorption rate and *t*_1/2_ (min) represents the adsorption half equilibrium time [45].

It can be seen clearly from the kinetic data in Figure 5 that the kinetic curve could be divided into two parts—the initial rapid adsorption and the subsequent mild equilibrium. The adsorption for uranyl ions reached 86.5% of the maximum adsorption capacity within 30 min, followed by the adsorption rate decreasing rapidly and attaining equilibrium. The obtained higher correlation coefficient values (*R*^2^) and the *Q_e_* much closer to the experimental value (Table 1) showed that the template binding kinetics of the micro-container could be better fitted by the pseudo-second-order model than another, which suggested that chemical adsorption acted as the rate-limiting step of the binding processes, just as in our previous research [45].

In order to study the dependence of the uranium binding properties on the initial concentration of UO_2_^2+^, the equilibrium adsorption capacity of the micro-container was investigated by performing batch adsorption experiments at a constant micro-container concentration and adsorption time but at a series of different initial concentrations of UO_2_^2+^ (*C_e_*). It can be seen clearly that the equilibrium adsorption capacity *Q_e_* increased as the *C_e_* increased, which was the common dependence of the equilibrium binding capacities of the micro-container on the initial adsorbate concentrations.

Two classical isothermal adsorption models, the Langmuir and Freundlich models, were then utilized to fit the above experimental binding data which is shown in Figure 6. The two fitted equations are empirical equations that represent two different binding behaviors. The Langmuir equation, shown in Equation (7), assumes monolayer adsorption on a finite number and identical sites, while the latter, shown in Equation (8), is based on multilayer adsorption on non-homogeneous sites. The adsorption constant *R_L_* in the Langmuir model is the basis for determining whether the experimental conditions for equilibrium adsorption are favorable or not, and its calculation equation is shown in Equation (9). When *R_L_* = 0, it means that the adsorption is irreversible; when 0 < *R_L_* < 1, it means that the adsorption conditions are favorable, when *R_L_* = 1, it means that the adsorption is linear in behavior; when *R_L_* > 1, it means that the conditions are unfavorable. In the Freundlich model, *K_F_* and *n* are constants that indicate the adsorption capacity and adsorption strength, respectively. If the value of 1/*n* is less than 1, it indicates that the Langmuir isotherm is normal; otherwise, it indicates that the adsorption is co-adsorption.
(7)$$Q_{e}=\frac{Q_{m}K_{L}C_{e}}{1+K_{L}C_{E}}$$
(8)$$Q_{e}=K_{F}C_{e}^{1/n},$$
where *Q_m_* (mg/g) is the maximum adsorption capacity. *K_L_* (L/mg) is the Langmuir adsorption constant. *K_F_* ((mg /mg) (L/mg)1/*n*) is the Freundlich adsorption constant and 1/*n* is the heterogeneity constant.
(9)$$R_{L}=\frac{1}{1+C_{m}K_{L}},$$
where *C_m_* (mg/L) is the maximum initial concentration of adsorption in the equilibrium adsorption experiment.

The two models’ fitted data are shown in Table 2. From the table, the *R*^2^ values of the Langmuir fitted model are higher than those of the Freundlich fitted model. So, the better fit of the Langmuir model suggests that the binding sites in the micro-container were homogeneous and indicates that the binding of the micro-container was a monolayer adsorption on the mean site with a maximum adsorption capacity of 111.9 mg/g. Moreover, the *R_L_* value was less than 1, also indicating that the condition of the micro-container was favorable for specific adsorption with uranium.

These results above confirmed that our design of the container-like block hydrogel was successfully achieved. The phosphorylated microgels were anchored into the macromolecule interpenetrating network to form the macro hydrogel, which was proved by several structure characterizations. Due to the existence of phosphate groups and the water compatibility of polymer skeleton, the micro-container showed a rapid adsorption rate and rather high adsorption capacity. Due to the macroscopic volume without the sacrifice of adsorption rate, this micro-container was expected to become a facile strategy for the emergency treatment of uranium leakage accidents.

### 3.3. Reflection Spectra of Micro-Container

Thanks to the self-assembly of PNIPAAm based microgels (VPA-microgel) in aqueous solutions, the structure of colloidal photonic crystals could be changed by the external environment which supplied the probability of the monitoring of the adsorption saturation process of the micro-container. After binding with uranium, the volume increased a little. Although the color changed from blue to white obviously (Figure 1c), the change of the reflection spectra of the micro-container before and after adsorption could be detected by using fiber optic spectroscopy. It can be seen clearly from Figure 7 that the reflection wavelength of the micro-container was changed from 572 nm to 637 nm after the saturation of adsorption. It was expected that the change of reflected light after the saturation of adsorbed uranyl ions was obvious enough to observe with the naked eye, which may provide a basis for the fast observation of whether the material was saturated with adsorbate or not.

## 4. Conclusions

The rapid, efficient and convenient extraction of uranium from water has been a top priority of human society in the nuclear industry. The super hydrophilicity and fast adsorption rate of adsorbents are always the main two aspects of adsorbent research and development. Moreover, the smart detection of adsorption saturation and a facile recovery strategy are better for controlling cost, the post treatment of adsorbent and the desorption of adsorbate. Our design of a uranium affine micro-container integrated the above features so that: (1) the PNIPAAm based polymer skeleton and the phosphate group provided a strong interaction with water and an affinity towards uranium, respectively; (2) 86.5% of UO_2_^2+^ was adsorbed after just 30 min, which is faster than other macro adsorbents [3]; (3) thanks to the structure of colloidal photonic crystals, the refection wavelength redshifted 65 nm after adsorption saturation, which was expected to indicate completion of adsorption with uranium detectable by the naked eye; (4) the hydrogel container grouped the microgels together by way of a macromolecule interpenetrating network, which was convenient for the recovery of adsorbent and uranium. Therefore, in this work, a novel micro-container with uranium affinity was developed in order to obtain both a high swelling rate and convenience of recycling, which will innovatively expand the strategies of uranium extraction.

## Figures and Tables

**Figure 1 nanomaterials-12-02259-f001:**
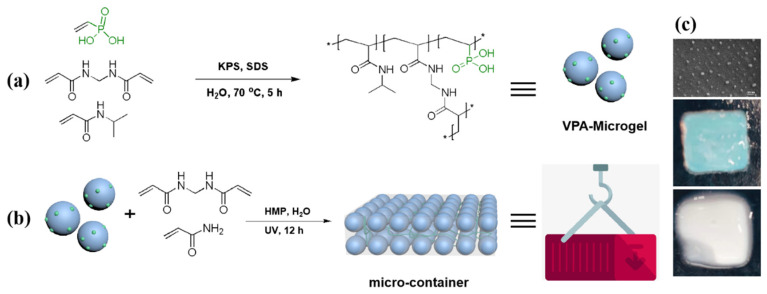
(**a**) The synthesis of the VPA-microgel. (**b**) Crosslinking of the preassembled VPA-microgel colloidal crystals via photoinitiated polymerization. (**c**) SEM photo of micro-container and the pictures of micro-container before (blue) and after (white) adsorption of uranium.

**Figure 2 nanomaterials-12-02259-f002:**
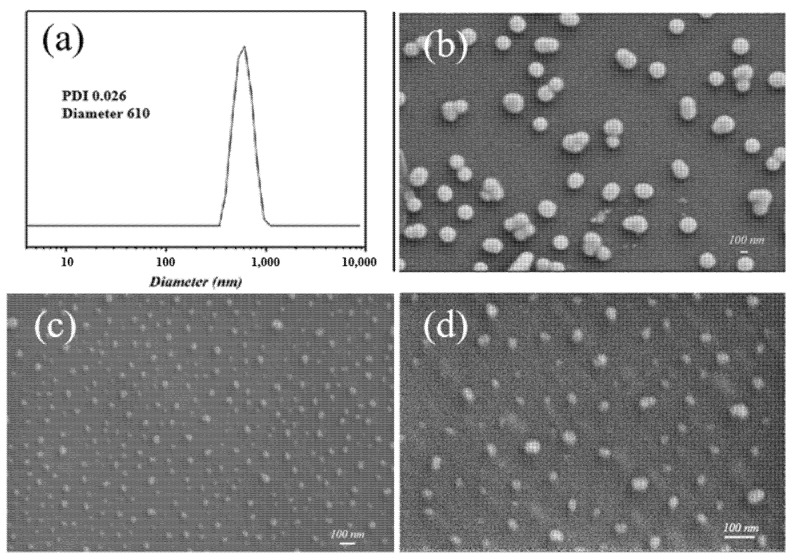
DLS diagram (**a**) and SEM image (**b**–**d**) of VPA-microgel (**a**,**b**) and micro-container (**c**,**d**).

**Figure 3 nanomaterials-12-02259-f003:**
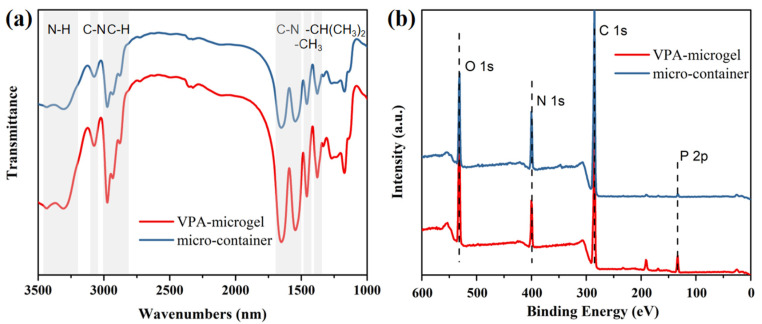
IR spectrum (**a**) and XPS diagram (**b**) of VPA-microgels (red line) and micro-container (blue line).

**Figure 4 nanomaterials-12-02259-f004:**
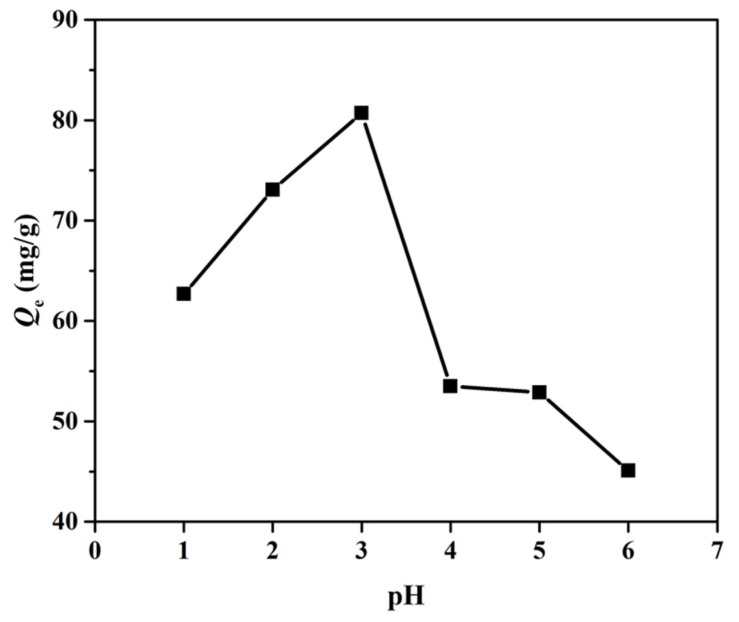
Effect of different acid concentration on adsorption capacity.

**Figure 5 nanomaterials-12-02259-f005:**
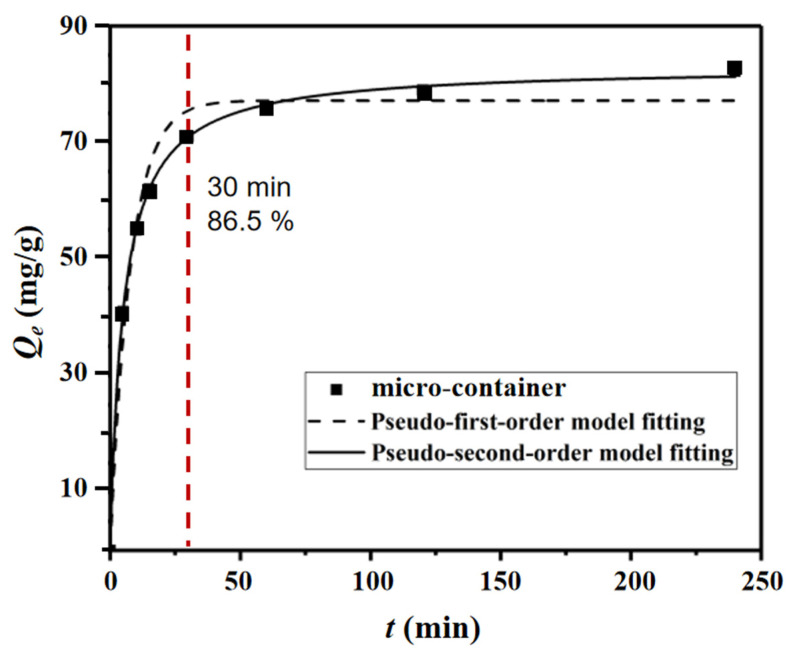
Kinetic data and modeling for the adsorption of uranyl ions onto micro-container.

**Figure 6 nanomaterials-12-02259-f006:**
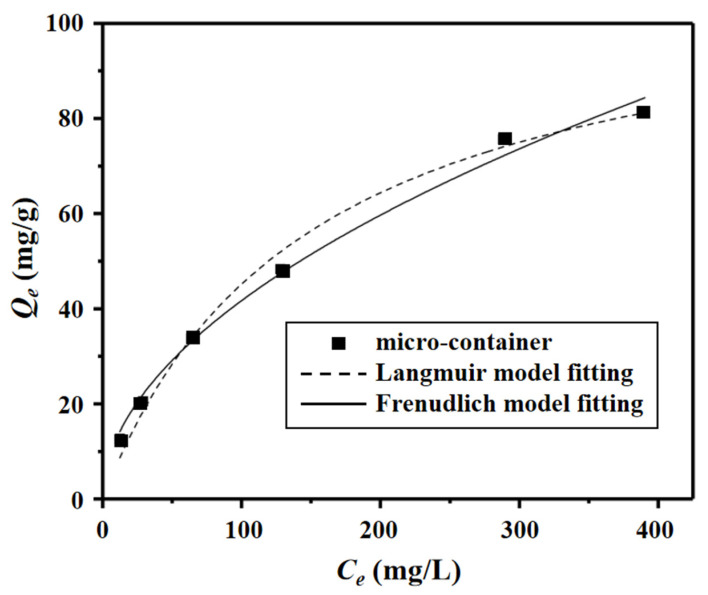
VPA-hydrogel equilibrium adsorption data and model fitting.

**Figure 7 nanomaterials-12-02259-f007:**
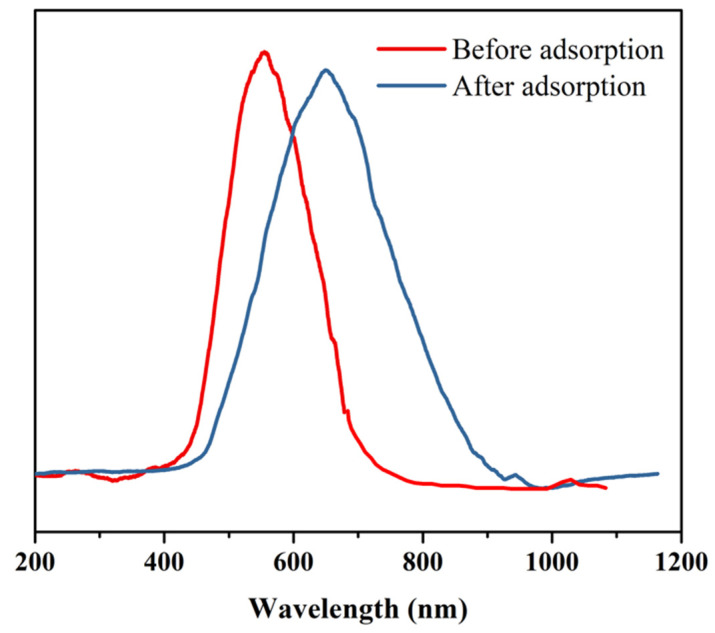
VPA-hydrogel reflection spectrum before and after adsorption.

**Table 1 nanomaterials-12-02259-t001:** Relevant parameters of micro-container kinetics adsorption.

	Pseudo-First-Order Model Fitting ^a^			Pseudo-Second-Order Model Fitting ^a^				
*Q_t_* (mg/g)	*Q_e_* (mg/g)	*k*_1_ (L/min)	*R* ^2^	*Q_e_* (mg/g)	*k*_2_ (g/(mg·min))	*R* ^2^	*h* (mg/(g·min))	*t*_1/2_ (min)
82.1	77.2	0.1284	0.966	83.1	0.0024	9.997	16.57	5.014

^a^*Q_e_* is the calculated binding capacities at equilibrium, *k*_1_ and *k*_2_ are the pseudo-first-order and pseudo-second-order rate constants of the binding processes, respectively, and *R*^2^ is the correlation coefficient value [3].

**Table 2 nanomaterials-12-02259-t002:** Binding isotherm constants of micro-container.

*T* (K)	Langmuir Model ^a^				Freundlich Model ^b^		
	*Q_m_* (mg/g)	*K_L_* × 10^2^ (L/min)	*R* ^2^	*R_L_*	*K_F_* (g/mg)·(L/mg)^1/*n*^	*R* ^2^	1/*n*
**273**	111.9	0.676	0.991	0.228	3.85	0.980	0.5173

^a^ *Q_m_* is the maximum adsorption capacity, *K_L_* is the Langmuir adsorption constant, *R_L_* is the adsorption constant; ^b^ *K_F_* is the Freundlich constant, *n* represents the adsorption strength and *R*^2^ is the correlation coefficient value.

## Data Availability

Not applicable.

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
