# Peer review of "The Efficient and Convenient Extracting Uranium from Water by a Uranyl-Ion Affine Microgel Container"

_nanomaterials, 2022, doi:10.3390/nano12132259_

Round 1
Reviewer 1 Report
The manuscript entitled "The Efficient And Convenient Extracting Uranium From Water By An Uranyl-Ion Affine Microgel Container“ by Pan Jianming et al. describes the synthesis and characterization of a novel nanomaterial, and its ability in uranium ions adsorption. The manuscript is well-written, and the presented results are interesting, however, the manuscript suffers from a serious methodological issue. The authors used uranium nitrate as a model compound to study the adsorption effectiveness of the nanomaterial. I see this as a problem for two reasons.
First, the uranium nitrate does not exist as UO22+ ion, but as neutral complex [UO2(NO3)2] (this is even more profound if nitric acid was used for adjustment of the pH). The bare uranyl(2+) ion exist in solution as a complex cation [UO2(H2O)4]2+. [UO2(H2O)4]2+ and [UO2(NO3)2] are chemically completely different species and thus the authors have to clarify which one do they really have present in their experimental setup. Moreover, due to even tiny changes in pH, another mixed aqua-, hydroxido-, and nitrato- complexes might be formed. If the experiment was not performed under an inert atmosphere, the formation of carbonato complexes such as [UO2(H2O)2(CO3)] might also take place since uranium(VI) exhibits a high affinity towards atmospheric carbon dioxide.
Second, if [UO2(NO3)2] is the actual species in the described experimental setup, it is not adequate to put this research into the context of uranium pollution, because in soil/biological systems the free U(VII) is present most likely in the form of aqua-, hydroxido-, and carbonato complexes.
Apart from this problem, the characterization of the new nanomaterial is reasonable, and the description and interpretation of the adsorption experiments is on a high level. Without clarification of the actual chemical species present in the solutions, however, it is not possible to compare the results with previously published data.
Author Response
Comment 1
First, the uranium nitrate does not exist as UO22+ ion, but as neutral complex [UO2(NO3)2] (this is even more profound if nitric acid was used for adjustment of the pH). The bare uranyl(2+) ion exist in solution as a complex cation [UO2(H2O)4]2+. [UO2(H2O)4]2+ and [UO2(NO3)2] are chemically completely different species and thus the authors have to clarify which one do they really have present in their experimental setup. Moreover, due to even tiny changes in pH, another mixed aqua-, hydroxido-, and nitrato- complexes might be formed. If the experiment was not performed under an inert atmosphere, the formation of carbonato complexes such as [UO2(H2O)2(CO3)] might also take place since uranium(VI) exhibits a high affinity towards atmospheric carbon dioxide.
Response 1:
Thank you very much for your very encouraging comments. According to the available literature, U(VI) ions exist in different forms at different pH, and in the adsorption conditions of this experiment at pH 3, U(VI) exists in the form of UO22+. ( J. Mol. Liq. 277 (2019) 843–855, J. Name., 2012, 00, 1-3)
Fig.1 Effect of ionic strength and effect of coexisting background electrolyte cations (J. Mol. Liq. 277 (2019) 843–855)
Comment 2
Second, if [UO2(NO3)2] is the actual species in the described experimental setup, it is not adequate to put this research into the context of uranium pollution, because in soil/biological systems the free U(VII) is present most likely in the form of aqua-, hydroxido-, and carbonato complexes.
Response 2:
Thank you very much for your critical advice. According to the reviewed data, U(VII) is rare in nature and U mainly occurs in +3, +4, +5, +6 valence states. This experiment is designed to treat U(VI) in wastewater. As mentioned above, at pH 3, U(VI) exists in the form of UO22+, so our material can be included in the treatment of uranium contamination.
Reviewer 2 Report
The presented article is interesting and a construction of article is logical. Scientific merit is good. The work is relevant and practical. Clarity of expression and communication of ideas, readability and discussion of concepts is medium. Article: The Efficient and Convenient Extracting Uranium from Water by a Uranyl-Ion Affine Microgel Container subject is very interesting and has practical use. The aim of this work was to investigate on Due to the rapid, efficient and convenient extracting uranium from water has been the top priority of human society in the field of nuclear industry. The super hydrophilicity and fast adsorption rate of adsorbent always are the two aspects of the adsorbent research and development. Moreover, the smart detection of adsorption saturation and the facile recovery strategy are better for the control cost, the post treatment of adsorbent and the desorption of adsorbate. Design of uranium affine micro-container integrated the above features that: 1) the PNIPAAm based polymer skeleton and the phosphate group provided the strong interaction with water and affinity towards uranium, respectively; 2) 86.5 % of UO2 2+ was adsorbed just after 30 min which is faster than other macro adsorbents; 3) thanks to the structure of colloidal photonic crystals, the refection wavelength redshifted 65 nm after adsorption saturation, which was expected to indicate completion of adsorption with uranium by naked eyes; 4) the container liked hydrogel grouped the microgels together by macromolecule interpenetrating network, which was convenient for the recovery of adsorbent and uranium. Therefore, in this work, the novel micro-container with uranium affinity was developed in order to obtain both high swelling rate and convenience of recycling, which will innovatively expend the strategies of uranium extraction.
However some corrections are needed:
1. It is necessary to emphasize the element of novelty in the publication.
2. Signs and symbols: Please, always use internationally accepted signs and symbols for units, SI units and give all dimensions according to the standard style of the Journal.3. If possible, the specific surface area of the adsorbent on which uranium adsorption occurs should be determined.
4. Was the specific activity in Bq or any radioactive dose tested, as is the case after the adsorption of uranium ions using this adsorbent, it would be very important and interesting information summarizing the publication.
4. Papers concerning The Efficient and Convenient Extracting Uranium from Water by an Uranyl-Ion Affine Microgel Container should be cited in Introduction section; for example:
Structural properties and adsorption of uranyl ions on the nanocomposite hydroxyapatite/white clay, Applied Nanoscience (Switzerland) 12 (4) (2022) 1101 – 1111
Adsorption of uranium ions on nano-hydroxyapatite and modified by Ca and Ag ions Adsorption 25 (2019) 639–647
Adsorption of Uranyl Ions at the Nano-hydroxyapatite and Its Modification Nanoscale Research Letters 12 (1) (2017) 278
Author Response
Comment 1.
It is necessary to emphasize the element of novelty in the publication.
Response 1:
Thank you very much for your critical advice. We have developed a uranium ion affinity and recoverable microgel container that not only has the rapid expansion rate of microgel particles, but also has much simpler recovery compared to microgels and has a naked eye detectable adsorption saturation process.
Comment 2.
Signs and symbols: Please, always use internationally accepted signs and symbols for units, SI units and give all dimensions according to the standard style of the Journal.
Response 2:
We have revised this according to your suggestion. We have changed mg/g to mg g-1 and mg/L to mg L-1 in the article.
Comment 3.
If possible, the specific surface area of the adsorbent on which uranium adsorption occurs should be determined.
Response 3:
Thank you very much for your very encouraging comments. We measured the specific surface area of the material after lyophilization as 749.8676 m² g-1.
Comment 4.
Papers concerning The Efficient and Convenient Extracting Uranium from Water by an Uranyl-Ion Affine Microgel Container should be cited in Introduction section。
Response 3:
Thank you very much for your suggestion. I have listed them in the cited literature [9,11].
Round 2
Reviewer 1 Report
The updated manuscript can be accepted for publication.